# Oxidation Behavior of the Monolayered La_2_Zr_2_O_7_, Composite La_2_Zr_2_O_7_ + 8YSZ, and Double-Ceramic Layered La_2_Zr_2_O_7_/La_2_Zr_2_O_7_ + 8YSZ/8YSZ Thermal Barrier Coatings

**DOI:** 10.3390/ma13143242

**Published:** 2020-07-21

**Authors:** Anna Jasik, Grzegorz Moskal, Marta Mikuśkiewicz, Agnieszka Tomaszewska, Sebastian Jucha, Damian Migas, Hanna Myalska

**Affiliations:** Department of Advanced Materials and Technologies, Silesian University of Technology, ul. Krasińskiego 8, 40-019 Katowice, Poland; anna.jasik@polsl.pl (A.J.); marta.mikuskiewicz@polsl.pl (M.M.); agnieszka.tomaszewska@polsl.pl (A.T.); sebastian.jucha@polsl.pl (S.J.); damian.migas@polsl.pl (D.M.); hanna.myalska@polsl.pl (H.M.)

**Keywords:** TBC, La zirconates, 8YSZ, degradation, TGO zone, mutual inner multiphase TBC

## Abstract

The degradation process of thermal barrier coatings (TBCs) such as monolayered La_2_Zr_2_O_7_, composite 50% La_2_Zr_2_O_7_ + 50% 8YSZ, and double-ceramic layer (DCL) La_2_Zr_2_O_7_/50% La_2_Zr_2_O_7_ + 50% 8YSZ/8YSZ was investigated. Coatings were deposited using the atmospheric plasma spraying (APS) process (ceramic layer and bond-coat) on the Ni-based superalloy substrate with Ni-22Cr-10Al-1Y bond-coat. The thickness of the ceramic top-coats in all cases were 300 µm. In the case of La_2_Zr_2_O_7_/8YSZ, the internal sublayer was built from 8YSZ powder whereas the outer from La_2_Zr_2_O_7_. Between both sublayers’ “composite” a 50% La_2_Zr_2_O_7_ + 50% 8YSZ zone was present. The “composite” 50% La_2_Zr_2_O_7_ + 50% 8YSZ TBC system was sprayed from two different feedstock powders with equal weight ratios. In the first part of the investigation, the microstructural characterization of the TBCs was presented. The main goals were related to the characterization of the degradation processes in different TBC systems with special emphasis on the phenomenon in the thermally grown oxide (TGO) zone related to oxidation, and the phenomenon related to phase stability in ceramic top-coats as related to temperature influence. The oxidation test was carried out in air at 1100 °C for 500 h. In the second step of the investigation, the numerical simulation of the monolayered TBC 8YSZ and La_2_Zr_2_O_7_ systems was analyzed from the stress distribution point of view. Additionally, the two-layered TBC coating of the DCL type was also analyzed.

## 1. Introduction

Thermal barrier coating (TBC) systems are the basic, and at present, essentially the only way to protect the surfaces of hot sections of gas turbine components such as combustion chambers or rotating blades in stationary units or aircraft engines [1]. Their application allows for the increase of the temperature of exhaust gases, and thus to increase efficiency [2]. The primary function of TBC systems is the thermal insulation of the substrate under conditions of an extreme working environment [3,4,5]. This effect can be obtained by the application of feedstock ceramic materials with low thermal conductivity, such as modified zirconia (e.g., 6–8 wt.% Y_2_O_3_ × ZrO_2_) or the rare-earth elements zirconates (with pyrochlore or fluorite type of lattice and an overall formula Ln_2_Zr_2_O_7_ where Ln = La, Sm, Nd, Gd, Yb, etc.). The effect of the thermal insulation can be strengthened by technological aspects such as the internal structure of coatings containing pores and special architecture of the cracks or interphase boundaries. Good examples of those solutions are thermal barrier coatings of a composite type [6,7,8,9] or double ceramic layer systems [10,11,12]. The concept of composite or DLC TBC systems application, in which a beneficial effect of thermal insulation appears with simultaneous parameters control, such as coefficient of thermal expansion (CTE), toughness, and Young’s modulus, is one of the most interesting plans to solve this problem. Additionally, it ensures a higher resistance to thermal shocks [6,7,8,9,10,11,12].

In such systems, the 8YSZ oxide is usually used as an inner ceramic layer in double-ceramic layer (DCL) systems, and it is characterized by CTE similar to bond coat, and by high thermo-chemical compatibility with Al_2_O_3_, which is a fundamental component of a thermally grown oxide (TGO) zone. Ceramic materials, characterized by better thermal properties, higher corrosion or erosion resistance, at usually simultaneous unbeneficial thermo-chemical compatibility with oxide of the TGO zone or value of CTE, constitute outer layers in insulation topcoat [13]. In the case of composite TBC systems, both compounds form a homogeneous ceramic layer with better insulation properties (as well as usually fracture toughness) than monolayered systems based on feedstock powders types 8YSZ and Ln_2_Zr_2_O_7_. The fundamental problems of the complex nature of the insulation layer can be related to mutual interaction between two different ceramic phases with the formation of new compounds, especially during high-temperature exposure in corrosive environments as well as the location of accelerated destruction processes in the area of Ln_2_Zr_2_O_7_/TGO interfaces with rapid oxide thickness growth [14]. 

Due to those phenomena, the most important aspect of internal morphology designing of insulation top-coat in a TBC system is recognition of:Degradation processes at the interface of the ceramic layer/TGO zone (8YSZ/Al_2_O_3_ or/and Ln_2_Zr_2_O_7_/Al_2_O_3_ for monolayered 8YSZ and Ln_2_Zr_2_O_7_ coatings, Ln_2_Zr_2_O_7_/8YSZDCL TBC‘ systems and composite Ln_2_Zr_2_O_7_ + 8YSZcoatings) related to oxidation;Interphase thermo-chemical stability between main components in composite systems, such as Ln_2_Zr_2_O_7_ + 8YSZ TBC, under conditions of oxidation and hot corrosion with assistance of liquid deposits; andDegradation of the top-coat insulation layer under the conditions of contact with corrosive environments such as liquid deposits of sulfate salts, vanadium oxides, or CMAS (deposits/8YSZ or/and deposits/Ln_2_Zr_2_O_7_ for monolayered 8YSZ and Ln_2_Zr_2_O_7_, Ln_2_Zr_2_O_7_/8YSZDCL and composite Ln_2_Zr_2_O_7_ + 8YSZcoatings) typical for hot corrosion processes.

The phenomena taking place in the area of the TGO zone related to the oxidation process were analyzed for monolayered La_2_Zr_2_O_7_ TBC as well as for the composite system type 50% La_2_Zr_2_O_7_ + 50% 8YSZ and DCL coatings with internal morphology, such as La_2_Zr_2_O_7_/50% La_2_Zr_2_O_7_ + 50% 8YSZ/8YSZ.

The conducted analysis gives the possibility of designing TBC systems with:An optimal value of thermal conductivity (presented in other publications [15]);High oxidation resistance (TGO zone morphology); andThermo-chemical stability between the 8YSZ and Ln_2_Zr_2_O_7_ phase (concept of mutual inner thermal barrier coatings [8,16]).

Highlighting the novelty of the presented paper:The microstructural characterization of the composite La_2_Zr_2_O_7_ + 8YSZ TBC system as a new solution for insulation layers—this new concept of the La_2_Zr_2_O_7_ + 8YSZ composite system was only described in three pieces of literature from 2018 to 2020. In [17,18,19], where the microstructural characterization, and some of the mechanical properties of this TBC system were presented. The investigation presented in the following publication is related to the microstructure (with special attention to the composite coatings‘ pores and cracks morphology descriptions) and the oxidation behavior of the dual phase La_2_Zr_2_O_7_ + 8YSZ system (layered and composite), as well as the phenomena in the TGO zone.The characterization of the composite system La_2_Zr_2_O_7_ + 8YSZ from the point of view of the phase stability and mutual interaction between both structural elements, as a very important factor in determining the overall durability of TBC—this element was only presented in Liu‘s team’s publications and was related to composite sinters [19,20,21,22,23], but there is lack of investigation for real TBC systems. In this investigation the mutual inertia in a plasma sprayed La_2_Zr_2_O_7_ + 8YSZ TBC system under high temperature oxidation test was analyzed, with special emphasis to the interface interactions in the area of the spalt-to-spalt boundaries. This zone is very important from the thermo-chemical point of view, and the durability of composite dual-phase TBC systems [18].

## 2. Experimental Procedure

All analyzed TBC systems were plasma sprayed using the APS method (atmospheric plasma spraying, F4MB plasma gun spraying by Metco, Oerlicon Metco, Westbury, NY, USA), with the use of commercial ceramic powders type 8YSZ (Sulzer Metco–M204NS) and La_2_Zr_2_O_7_ (Trans-Tech). The overall morphology of those powders (SEM—scanning electron microscopy -Hitachi 3400N, Hitachi, Tokyo, Japan) and their chemical and phase composition analyzes were conducted using ICP-OES (inductively coupled plasma optical emission spectrometer-Ultima 2 ICP OES, Horiba Scientific, Tokyo, Japan), main elements: HFIR (high-frequency infrared, Coulomat 702 Strohlein, Ströhlein-Labexchange Service GmbH, Burladingen, Germany), analysis of carbon and sulfur: high-temperature extraction (ON-mat 8500 Strohlein), analysis of oxygen and nitrogen: XRD (X-ray diffraction, X‘Pert^3^ Powder, Malvern Pananalytical, Malvern, UK), phase composition and EBSD (electron backscattered diffraction, Hitachi 3400N with INCA HKL Nordlys II (Channel 5)), phase constituents in micro-areas are shown in Table 1 and in Figure 1, Figure 2 and Figure 3. The main technological parameters of the deposited powders such as particle size distribution (laser diffraction, Mastersizer with Hydro 2000S by Malvern Inc., Malvern, UK), density (PN-EN ISO 18753, PN-EN 23923-1, and PN-EN 23923-2), and flowability (PN-EN ISO 4490) are presented in Figure 4 and Table 2. Those data revealed similar technological properties of the powders used.

The chemical composition of the analyzed powders applied in the TBC systems depositions revealed a high quality of feedstock material. Especially the high level of phase constituent homogeneity of the zirconate powder, containing mainly La_2_Zr_2_O_7_ phase with a pyrochlore type of lattice. No other phases were detected by XRD and EBSD analysis. In the case of the 8YSZ powder typical phase composition containing dominant tetragonal phase a small amount of monoclinic zirconia was found. The cubic phase of the ZrO_2_ type can be present in the powder as well, but XRD and EBSD analysis did not reveal this phase. The data obtained revealed a high level of similarity between both feedstock materials. It can suggest that the APS process can provide the correct course without the effect of powders segregation in the case of composite and DCL coatings depositions.

The parameters of the APS process for each type of TBC system are listed in Table 3. The In 625 Ni-based superalloy was used as a substrate material with a bond-coat of 125 μm thickness obtained from Ni22Cr10AlY (−106 + 53 μm particle size, Oerlikon Metco, Amdry 962). The bond-coat was sprayed using the APS method as well. A monolayered 8YSZ coating was used as reference material. The 50% La_2_Zr_2_O_7_ + 50% 8YSZ composite and double ceramic layered La_2_Zr_2_O_7_/50% La_2_Zr_2_O_7_ + 50% 8YSZ/8YSZ DCL coatings were sprayed with the use of two separate powders feeders. The thickness of the ceramic top-coat was in the range from 250 to 300 µm.

The microstructure of all (monolayered, composite, and double ceramic layered) obtained TBC systems are shown in Figure 5. The microstructure observations of TBC’s systems confirm the correct process of coating deposition considering especially the homogeneity of the coatings (no segregation of either phases was observed). The quality of the ceramic top-coat is very high with the porosity absent massive effects (overall integrity, uniform thickness, globular particles, horizontal cracks, contaminations, unmelted particles, interphases cracks between bond-coat and top-coat, adhesion to the substrate, etc.).

The thickness of the TBC systems was in a range of 225–300 µm (225 µm–monolayered La_2_Zr_2_O_7_, 250 µm-DCL La_2_Zr_2_O_7_/50% La_2_Zr_2_O_7_ + 50% 8YSZ/8YSZ, and 300 µm composite TBC La_2_Zr_2_O_7_ + 8YSZ. A more detailed microstructural analysis of the analyzed TBC systems was shown in [8,15]. The presented investigation revealed that the composite system is characterized by a much higher level of porosity (ca. 18%), than the DCL (14%) and monolayered TBC (11%). This difference is related to the different value of the La_2_Zr_2_O_7_ + 8YSZ melting point, much higher for 8YSZ.

In consequence the morphology of pores should be different in dual phase systems in comparison to monophasic TBC. This information was confirmed by detailed image analysis of the pores and cracks. In the case of the monolayered system, dominating forms of voids are spherical pores with small size, but their surface fraction is much smaller than for non-spherical voids (mainly cracks)—Figure 6. The highest surface fraction of non-spherical voids was observed for the composite systems, where pores have strongly complexed shapes—Figure 7. In the case of the DCL coating the morphology of pores is indirect—Figure 8.

The monolayered coatings of 8YSZ and La_2_Zr_2_O_7_, composite TBC La_2_Zr_2_O_7_ + 8YSZ, and DCL TBC La_2_Zr_2_O_7_/50% La_2_Zr_2_O_7_ + 50% 8YSZ/8YSZ were examined under the conditions of oxidation testing at a temperature of 1100 °C and the total time of exposure was 500 h. The microstructure, chemical composition (SEM/EDS—scanning electron microscopy, energy dispersion spectroscopy—Hitachi 3400N with Thermo NORAN System Seven) and phase composition (XRD- X’Pert^3^ Powder) of the corrosion products were evaluated in all cases. In the case of the oxidation test, the phenomena occurring in the TGO zone were also analyzed by the LM (light microscopy-Olympus DP70), SEM/EDS (Hitachi 3400N with Thermo NORAN System Seven) and XRD/EBSD (Hitachi 3400N with INCA HKL Nordlys II (Channel 5)) methods. The oxidation behavior of the 8YSZ monolayered TBC systems are widely described in the literature and the data are presented in this article only as reference material. 

In order to analyze the failure mode related to stress distribution in the area of La_2_Zr_2_O_7_/TGO and 8YSZ/TGO interfaces, finite element simulation was performed. The Algor program was used to carry out the computer simulation. With the complexity of the calculation procedures, a method of minimizing the task by using axial-symmetrical calculation models was presented, which made it possible to solve half of the task (on the axis the boundary condition resulting from the symmetry: displacement in the Y = 0 direction). 2D axial-symmetrical and four-node elements were adopted in the analyzed problem. The task is solved with contact and friction. Boundary conditions were set: the displacement of the lower surface of the model in the direction of the *z*-axis was removed. The corresponding material parameters have been assigned to the model [24].

The monolayered TBC model with a ceramic layer of total thickness of 225 µm was subjected to numerical analysis. The thickness of the bond-coat was 125 µm. The first stage of the research was to conduct a numerical analysis of the temperature field. The conditions of the heat exchange on the external surfaces of the body were determined by setting boundary conditions. They can be determined for the conduction of heat in solids in different ways. In the analyzed model, the boundary conditions of the first type, the so-called Dirichlet conditions, determined by temperature distribution T_S_ on the solid surface at any time, were set. Considering temperature fields in solid states, the solution of the problem was formulated as a simple task, consisting of the determination of temperature distribution from given initial and boundary conditions as well as material properties of the considered body. 

The temperature load on the surfaces of the model had been taken into account. On the surface of the ceramic layer a temperature of 1200 °C was applied, while on the external surface of the nickel superalloy, a temperature of 800 °C was adopted. Those analyses were the basis of the stress-strain evaluation in the monolayered TBC systems. The basis of the calculations in this case is a numerical calculation model used in the thermal calculations in conjunction with information about the temperature values at each geometric point of the object. This is a prerequisite for the definition of the boundary conditions in the strength analysis. The critical stress-strain state with varied bond-coat thickness and loaded with a constant temperature field was evaluated. The freedom of the models was restricted. It was assumed that the cross sections of the developed models do not move in the direction of the *z*-axis. Knowing the axial-symmetric temperature field in the analyzed models, the stress state present in them was determined in the analyzed areas.

## 3. Results and Discussion

In the first stage of the investigation, an oxidation test at a temperature of 1100 °C in laboratory oxygen for 500 h was carried out. The TBC samples of La_2_Zr_2_O_7_, 50% La_2_Zr_2_O_7_ + 50% 8YSZ, and La_2_Zr_2_O_7_/50% La_2_Zr_2_O_7_ + 50% 8YSZ/8YSZ were collected after 2, 10, 48, 175, and 500 h of exposure. This analysis enabled the evaluation of the overall quality of the top-coat of the ceramic layers after the test and macroscale destruction of the TBC systems. The exemplary photos of all TBC systems are presented in Figure 9, where La_2_Zr_2_O_7_, 50% La_2_Zr_2_O_7_ + 50% 8YSZ and La_2_Zr_2_O_7_/50% La_2_Zr_2_O_7_ + 50% 8YSZ/8YSZ systems are shown after 48, 500, and 500 h of static oxidation test, respectively.

The visual evaluation shows unequivocally much better behavior of the composite and DCL systems in comparison to the monolayered La_2_Zr_2_O_7_ coating, which was destroyed after 48 h of exposure. The 50% La_2_Zr_2_O_7_ + 50% 8YSZ and La_2_Zr_2_O_7_/50% La_2_Zr_2_O_7_ + 50% 8YSZ/8YSZ systems preserved their integrity until the end of the test after 500 h. It should be noted that the thickness of the composite and the DCL TBC systems is much higher than in the case of the monolayered coatings (a higher thickness of ceramic layer usually leads to a higher tendency to spallation). Despite this, their behavior is much more beneficial than expected.

The macroscopic effects are related to the spallation of the ceramic insulation layer expressed by so-called “white failure” of the La_2_Zr_2_O_7_ coating. As a consequence, the main part of the insulation layer was removed from the surface and the protection is provided by the thin internal part of the ceramic top-coat. This process accelerated the oxidation of the bond-coat surface and the rapid increase of TGO thickness, which is visible in Figure 10, where the strong acceleration of the TGO zone thickness was observed after 48 h of exposure (kinetic of oxidation very close to parabolic). Those effects are controlled by rapid oxygen diffusion across the residual ceramic insulation layer. In the case of the composite and DCL coatings, where the interphase zone of the ceramic top-coat/bond-coat is characterized by the reaction types of 50% La_2_Zr_2_O_7_ + 50% 8YSZ/Al_2_O_3_ and 8YSZ/Al_2_O_3_, the zone growth is strongly lower and mutually similar. 

The most beneficial effect was observed for reference to the 8YSZ TBC system, where the TGO zone thickness was the lowest. Observed differences in the TGO growth kinetic are related to oxygen transport across the ceramic top-coat which is much more effective (unbeneficial effect) for strongly defected pyrochlore compounds such as La_2_Zr_2_O_7_. The same effect provides exceptionally good insulation properties (very beneficial effect) related to phonon scattering susceptibility. The higher level of oxygen transferred to the surface of Ni_22_Cr_10_AlY bond-coat generated the changes in TGO zone morphology such as the formation of a thicker layer of Al_2_O_3_, additionally rich in spinel oxides such as (Ni,Cr)Al_2_O_4_ or complex oxides of the LaAlO_3_ type with a perovskite-type of a lattice. It should also be noted that the thickness of the monolayered system was the lowest of those investigated (the highest TGO thickness), but on the other side, the porosity level, and non-spherical pores fraction in total porosity, was the most undesirable in the case of the composite TBC, and DCL systems (strongly developed system of oxygen-transport channel).

Examples of the TGO zone morphology identified for three different types of TBC systems are shown in Figure 11, Figure 12, Figure 13, Figure 14, Figure 15, Figure 16, Figure 17, Figure 18, Figure 19, Figure 20, Figure 21, Figure 22, Figure 23 and Figure 24. The most crucial difference is the thickness of the oxide zone as related to the formation of the multiphase and at least two-layered morphology of the TGO zone. The TGO zone is generally formed from the Al_2_O_3_ oxide, which is a dense layer with a columnar structure. The thickness of this sub-zone is stable and homogeneous both in the top and bottom regions of bond-coat (related to technological roughness beneficial from an adhesion point of view). This morphology is observed for the TBC systems with the internal part of the insulation layer, built from 8YSZ ceramic (8YSZ monolayered TBC or La_2_Zr_2_O_7_/50% La_2_Zr_2_O_7_ + 50% 8YSZ/8YSZ DCL TBC). When the oxygen concentration was relatively high, the additional porous sub-layer consisting mainly of spinel-type of oxides such as (Ni,Cr)Al_2_O_4_ was formed. The thickness of this sub-zone is much higher and nonhomogeneous, especially when the data from the top and bottom regions of the bond-coat are compared. 

This morphology is observed especially for the monolayered La_2_Zr_2_O_7_ TBC. Additionally, in this case, the transition oxides such as LaAlO_3_ can be formed, but they do not play an important role in the degradation process related to “white failure” and the spallation of ceramic top-coat. The presence of perovskite-type LaAlO_3_ was confirmed by an EBSD analysis (Figure 23). In the TBC systems with an internal layer based on 8YSZ, the main identified compounds were Al_2_O_3_ and NiAl_2_O_4_. Those phases were also dominant compounds in the TGO zone of the DCL type of coating. The presence of the LaAlO_3_ phase was confirmed only locally. The analysis of the La_2_Zr_2_O_7_ + 8YSZ composite coatings revealed intermediate behavior related to the formation of the TGO zone with a thickness slightly higher in comparison to the DCL coating and phase composition based on Al_2_O_3_ and NiAl_2_O_4_. The presence of perovskite-type oxides was not detected.

The morphology of the TGO zone (expressed by phase composition, and the thickness of the oxide scale is the most important factor in determining the TBC life-time during the oxidation condition. Generally, the critical thickness of the TGO zone, associated with the start of crack nucleation and growth with the final spallation effect, is determined at a level about 6 to 8 µm for Al_2_O_3_ based TGO. An additional effect is related to the inhomogeneity of the TGO thickness, especially in the area of bond-coat top areas, which was shown in Figure 25. The average value of the oxides zone is lower than the critical value, but locally, the thickness is frequently higher. For example, in the case of monolayered La_2_Zr_2_O_7_ TBC, the average value of the TGO zone is 4.9 µm, but locally this zone had the thickness ca. 25 µm. This effect probably strongly accelerated the destruction process and the effect of ceramic layer spallation.

The observed effects of crack formation (in the ceramic insulation layer in areas above the TGO zone, especially directly above the peaks of bond-coats profile roughness—Figure 26a) are caused by unbeneficial stress distribution related to the relatively high value of the elastic modulus of La zirconates. As a consequence, the generated stress is not compensated by the porous structure of plasma sprayed coatings, and finally spallation of the whole ceramic layer—so-called “white failure”—can be observed. This assumption was confirmed by numerical simulation by FEM (finite elements modelling) for the monolayered TBC systems of La_2_Zr_2_O_7_ type, shown at Figure 27, where the stress level on the cross-sectioned modelled TBC system, clearly showed that the most beneficial area for crack formation is located directly above the TGO zone. The visual effect of spallation was detected after 48 h of exposure at a temperature of 1100 °C. The detailed analysis of the TGO zone revealed, in this case, that there were no microstructural factors which can generate strong impulse for this drastic type of destruction.

The effect of dramatic changes in the stress value was not observed for the DCL coatings, where the internal zone is built from 8YSZ ceramic. The FEM simulation of the stress distribution for 8YSZ based TBC systems revealed that for this type of coating, the stress distribution on the interface between the ceramic top-coat and bond-coat is very beneficial, without drastic changes of their value and type. These assumptions were also confirmed by microstructural investigation presented in Figure 26b, where no effect of spallation was detected. In the case of both TBC systems, the TGO zone was characterized only by the formation of an oxide scale built from the dense Al_2_O_3_ and porous NiAl_2_O_4_. The Al_2_O_3_ phase was dominant in the 8YSZ TBC, and the spinal phase was the main structural element of the TGO zone in the La_2_Zr_2_O_7_ TBC coating. Additionally, the effects of relatively strong NiAl_2_O_4_ based scale growth was detected for the La_2_Zr_2_O_7_ based TBC, with penetration of ceramic top-coat.

The SEM investigation revealed the formation and growth of small perpendicular cracks penetrating the zirconate phase, and as the final stage, the extraction of micro-fragments of La_2_Zr_2_O_7_ splats. A similar effect, but at a much lower scale (lower fraction of spinel oxides) was also observed for the DLC coating with an 8YSZ phase as an element of interface with the TGO zone. It suggests that the TBC systems of a DCL type (in this case with an additional composite sublayer of La_2_Zr_2_O_7_/50% La_2_Zr_2_O_7_ + 50% 8YSZ/8YSZ type) are characterized by much beneficial stress distribution.

Another factor (important for multiphase TBC systems such as composite and DCL La_2_Zr_2_O_7_ + 8YSZ and La_2_Zr_2_O_7_/50% La_2_Zr_2_O_7_ + 50% 8YSZ/8YSZ, respectively) is thermo-chemical interphase stability in the dual-phase system. The lack of mutual stability may lead to decomposition of one or both phases or its mutual dissolution.

The XRD analysis of the monolayered TBC system, with outer La_2_Zr_2_O_7_ sublayer, showed that this pyrochlore phase is stable at least to 1100 °C during exposure for 500 h. The XRD patterns (Figure 28) showed this stability. That is the starting point for the analysis of the two-compound system of La_2_Zr_2_O_7_ + 8YSZ type. Eventually, observed phase instabilities must be related to mutual dissolutions or decomposition of one or both phases where the coexisting phase is the catalyst of this process. The analysis presented for composite systems of Sm_2_Zr_2_O_7_ + 8YSZ types revealed the possibility of this phenomenon occurring [16], where the effect of the pyrochlore Sm_2_Zr_2_O_7_ phase decomposition to the fluorite type was observed, as an effect of high-temperature exposure at a temperature of 1100 °C for 500 h of exposure.

In the analyzed case of the La_2_Zr_2_O_7_ + 8YSZ TBC system, a reaction between the two phases was not detected, which confirmed the XRD data for 50% La_2_Zr_2_O_7_ + 50% 8YSZ TBC system after 500 h of exposure at 1100 °C (Figure 29). The SEM analysis of this TBC system did not confirm the phenomena related to mutual interaction between both phases (Figure 30).

Analysis of the top-coat surface of all TBC systems in the as-sprayed condition as well as after 2, 10, 48, 175, and 500 h of exposure at 1100 °C revealed changes in ceramic layer topography. In the case of the as-sprayed condition, the splats’ morphology is typical for the APS process with remelted, larger oval granules with internal cracks as a consequence of rapid cooling and a high level of internal stress occurring after deposition. In the case of smaller granules, their shape is similar to droplets stationed on the surface.

The same morphology of topcoat was observed after 2 and 10 h of exposure. After 48 h of exposure, significant differences can be observed such as rough surface, sharp edges of the splats, and increasing levels of voids on the splats boundaries which finally form the network cracks on the surface of the topcoat after 175 and 500 h of exposure. Additionally, the network of primary cracks (results of splats crystallization during the spraying process) disappeared. Those effects suggest that the sintering process was a significant factor influencing the surface condition of the TBC systems (Figure 31).

## 4. Summary

The presented investigation revealed an extraordinarily strong influence of the internal morphology of the insulation layer on the overall durability of the TBC systems in operational conditions. One of the most important destruction modes is oxidation behavior, which is controlled only by phenomena in the TGO zone. At this moment, there is no better method than decreasing TGO zone growth kinetics by modification of the chemical composition of the bond-coat with Al_2_O_3_ formation as a primary goal of those changes. Under this condition, the most critical factor related to TGO morphology is thermo-chemical stability between alumina and zirconia, as a material of the ceramic insulation layer based on 8YSZ. 

The development of new ultra-low conductivity materials based on zirconate rare earth elements gives new possibilities to new morphological conceptions of TBC systems. The thermal conductivity can be strongly limited by applications of pyrochlore materials of La_2_Zr_2_O_7_ types (stronger effect of phonon scattering). Additionally, the new ceramic materials exhibit better corrosion resistance in liquid deposits, but due to a high level of internal defects, the ion conductivity is the reason for higher susceptibility to oxygen diffusion through the ceramic layer. As a consequence, a higher amount of oxygen is transferred to bond-coats, and the TGO zone reveals undesirable high kinetic growth. The beneficial influence of zirconates such as La_2_Zr_2_O_7_, inducting the improvement of insulation properties and better behavior under conditions of hot corrosion, with undesirable decreasing of oxidation resistance is the reason for new concepts in internal morphology of TBC systems taking into consideration both beneficial and undesirable effects. 

The most promising solutions are composite La_2_Zr_2_O_7_ + 8YSZ and double ceramic layered La_2_Zr_2_O_7_/La_2_Zr_2_O_7_ + 8YSZ/8YSZ TBC systems that were analyzed in this article. Analysis of thermal conductivity (presented in other investigations [15,25]) showed clearly that multiphase internal structures (both in composite type and DCL TBC as well) enhanced the phonons scattering effect by the presence of interphases (intersplats) boundaries, and thermal resistance of those types of coatings is much higher than for monolayered based on La_2_Zr_2_O_7_ with the same thickness. Additionally, the TGO growth kinetic is strongly limited in composite coatings, and especially DCL coatings in comparison to monolayered zirconate based systems. The absolute value of TGO zone thickness is similar to that obtained for conventional monolayered 8YSZ TBC. It can be assumed that analysis of phenomena in the TGO zone showed clearly that the most beneficial effect could occur only when an internal sublayer in multiphase TBC systems consists only of the 8YSZ phase, which is very stable thermo-chemically in contact with Al_2_O_3_ from the TGO zone. Another phenomenon rarely described in literature is the thermo-chemical stability of interfaces in the La_2_Zr_2_O_7_ + 8YSZ system (for composite and DCL coatings). This element is a new factor important for overall durability from the TBC systems point of view, and can be the reason for phase instability in the ceramic layer. 

As a consequence the effect of ceramic insulation layers can be observed. This phenomenon was observed in the case of the composite Sm_2_Zr_2_O_7_ + 8YSZ TBC systems, and was expressed by the pyrochlore phase decomposition with the formation of fluorite type of samarium zirconates [16]. Additionally, the strong effect of selective corrosion of Sm_2_Zr_2_O_7_ was observed.

It can be assumed that achievements of better insulation properties by composite or DCL type TBC have been replaced by a new generation problem related to interphase thermo-chemical stability in multiphase TBC systems of the new type. This is an important design problem that was shown during investigations of Sm_2_Zr_2_O_7_ + 8YSZ composite coatings, where the effect of decomposition was strong. Additionally, selective corrosion and structural destruction of the pyrochlore phase were also observed [16]. The literature data [26,27,28,29] describes generally the effect of thermal decomposition of single-phase TBC systems of La_2_Zr_2_O_7_. However, the presented problem showed that mutual chemical inertness between phases co-existed in TBC systems such as 8YSZ and different types of rare earth zirconates should be explained, and a conception of mutual inner thermal barrier coatings (MITBC) is a good direction of investigation. Presented in this article, results can suggest that a system of type La_2_Zr_2_O_7_ + 8YSZ can meet these expectations. 

## 5. Conclusions

The investigation presented in the article suggests that the most beneficial effect related with the overall durability of TBC systems in high-temperature oxidation condition was achieved by the double-layered ceramic TBC system with the internal sub-layer built from 8YSZ powder, the outer from lanthanum zirconate La_2_Zr_2_O_7,_ and the transition constructed from a composite mixture of both analyzed phases La_2_Zr_2_O_7_ + 8YSZ.

The internal sub-layer based on 8YSZ is responsible for oxidation behavior and a decrease of TGO zone growth, and it ensures more beneficial stress distribution, which effectively inhibits crack formation and spallation of the ceramic top-coat.

The presence of composite transition layer in DCL coating is next to mentioned earlier 8YSZ inner layer, the reason of beneficial behavior of DCL TBC system of La_2_Zr_2_O_7_/50% La_2_Zr_2_O_7_ + 50% 8YSZ/8YSZ type under the condition of high-temperature oxidation. This observation is confirmed by the analysis of composite TBC systems of the 50% La_2_Zr_2_O_7_ + 50% 8YSZ type, where no effect of spallation of the ceramic top-coat was observed.

The results obtained revealed that the La_2_Zr_2_O_7_ + 8YSZ system is thermo-chemically stable in a condition of high-temperature oxidation and can be considered as dedicated for mutually inner thermal barrier coatings (MITBCs) designs.

## Figures and Tables

**Figure 1 materials-13-03242-f001:**
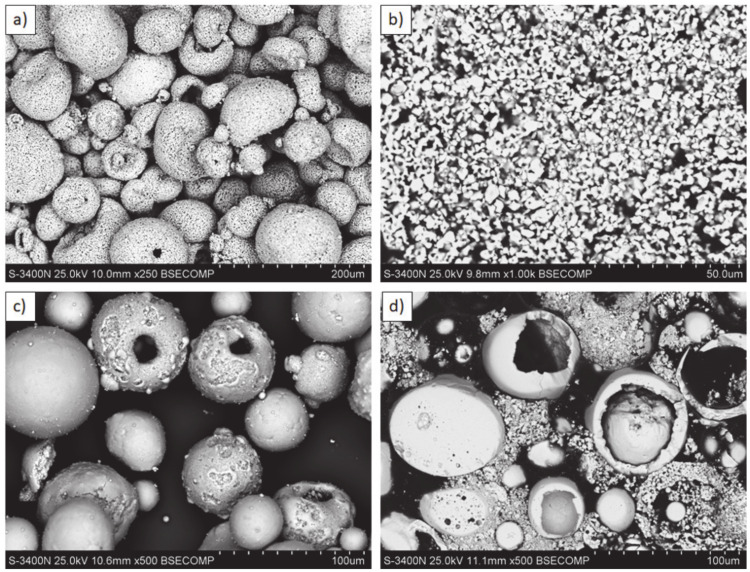
Feedstock powders: (**a**,**b**) La_2_Zr_2_O_7_: morphology and cross section; (**c**,**d**) 8YSZ: morphology and cross section.

**Figure 2 materials-13-03242-f002:**
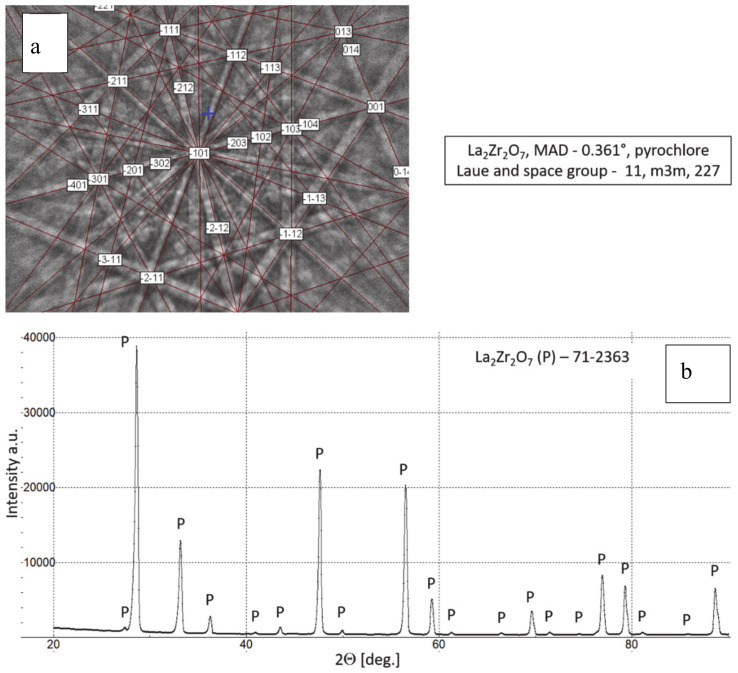
Phase composition of feedstock La_2_Zr_2_O_7_ powder, (**a**) Kikuchi lines: EBSD (electron backscattered diffraction) method and (**b**) XRD (X-ray diffraction) pattern.

**Figure 3 materials-13-03242-f003:**
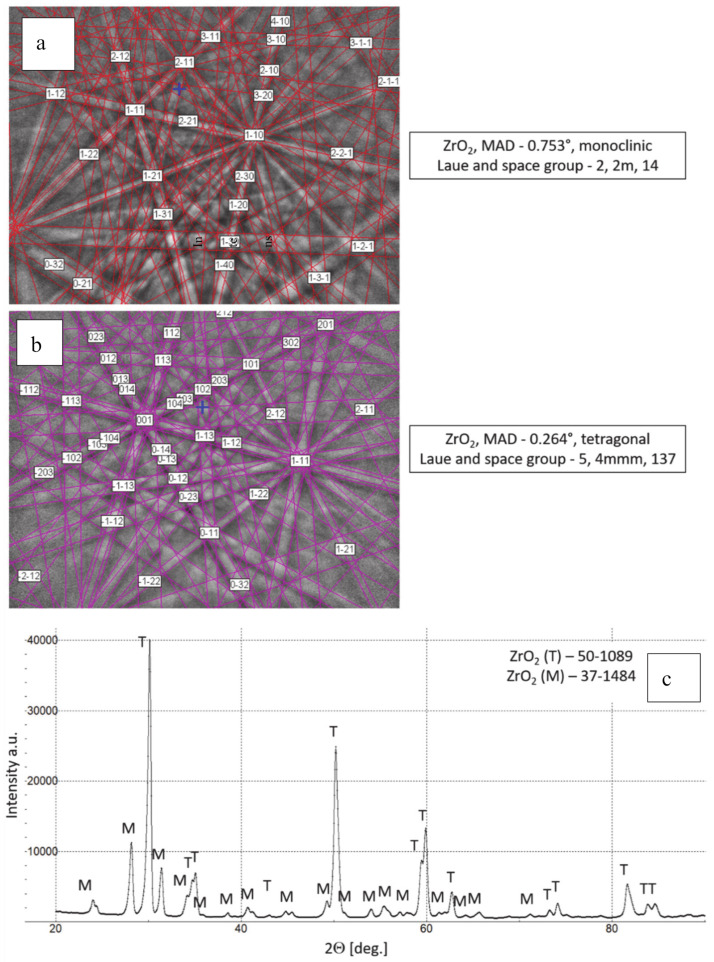
Phase composition of feedstock 8YSZ powder, (**a**,**b**) Kikuchi lines-EBSD method, (**c**) XRD diffraction pattern.

**Figure 4 materials-13-03242-f004:**
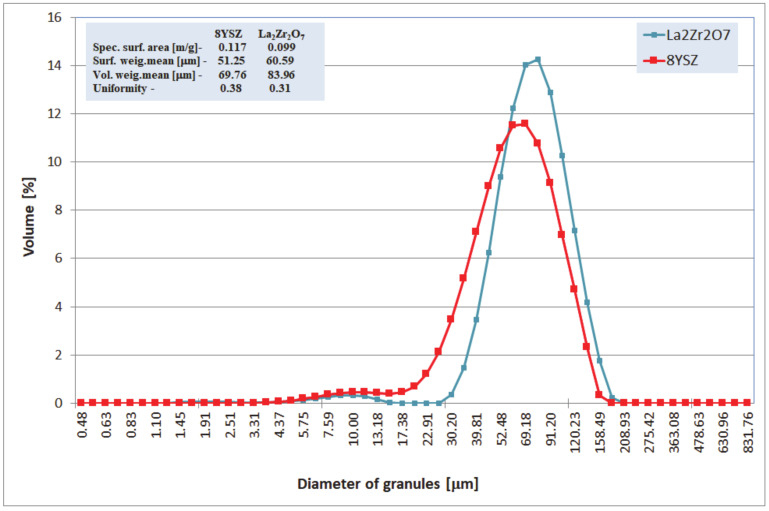
Characterization of La_2_Zr_2_O_7_ and 8YSZ powder size distributions.

**Figure 5 materials-13-03242-f005:**
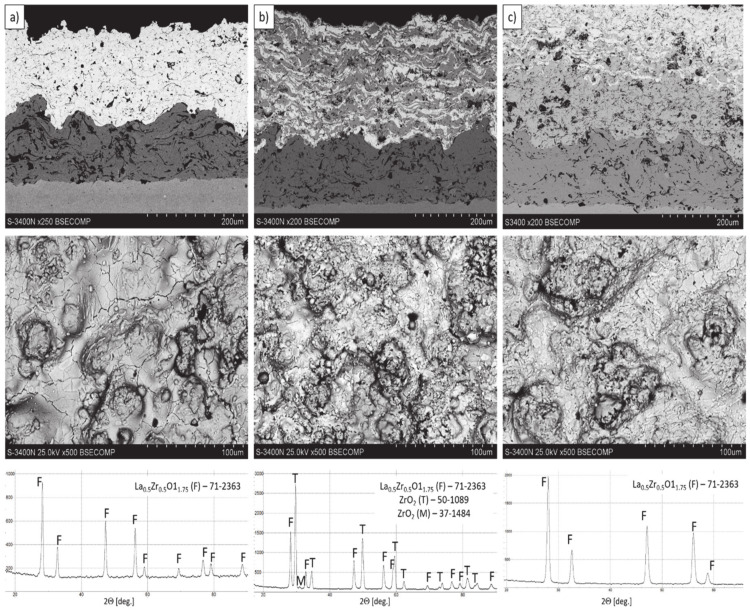
The cross-section microstructure, surface topography, and phase composition of TBC systems: (**a**) monolayered La_2_Zr_2_O_7_, (**b**) composite 50% La_2_Zr_2_O_7_ + 50% 8YSZ, and (**c**) DCL La_2_Zr_2_O_7_/50% La_2_Zr_2_O_7_ + 50% 8YSZ/8YSZ.

**Figure 6 materials-13-03242-f006:**
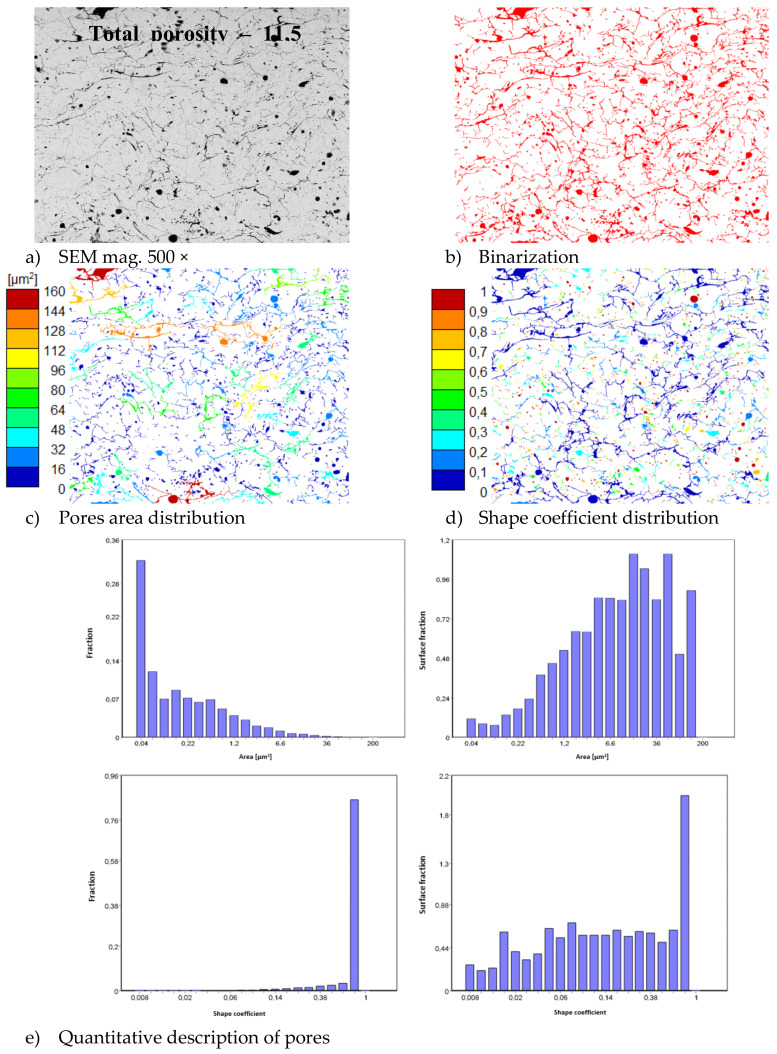
The image analysis results describe pores and voids morphology from quantitative and qualitative points of view for the monolayered La_2_Zr_2_O_7_ system: (**a**) SEM image, (**b**) binarization of pores and cracks, (**c**) visual evaluation of pores area distribution, (**d**) visual evaluation of pores shape coefficient distribution, (**e**) quantitative evaluation of pores area and shape coefficient distribution.

**Figure 7 materials-13-03242-f007:**
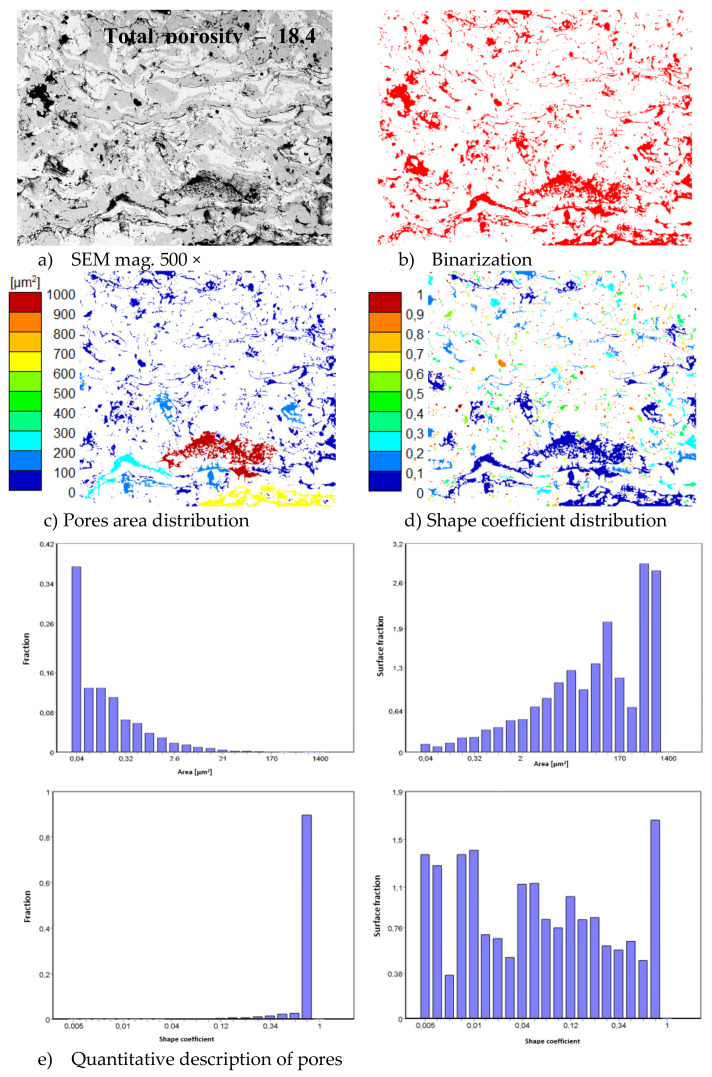
The image analysis results describe pores and voids morphology from quantitative and qualitative points of view for the composite 50% La_2_Zr_2_O_7_ + 50% 8YSZ system: (**a**) SEM image, (**b**) binarization of pores and cracks, (**c**) visual evaluation of pores area distribution, (**d**) visual evaluation of pores shape coefficient distribution, (**e**) quantitative evaluation of pores area and shape coefficient distribution.

**Figure 8 materials-13-03242-f008:**
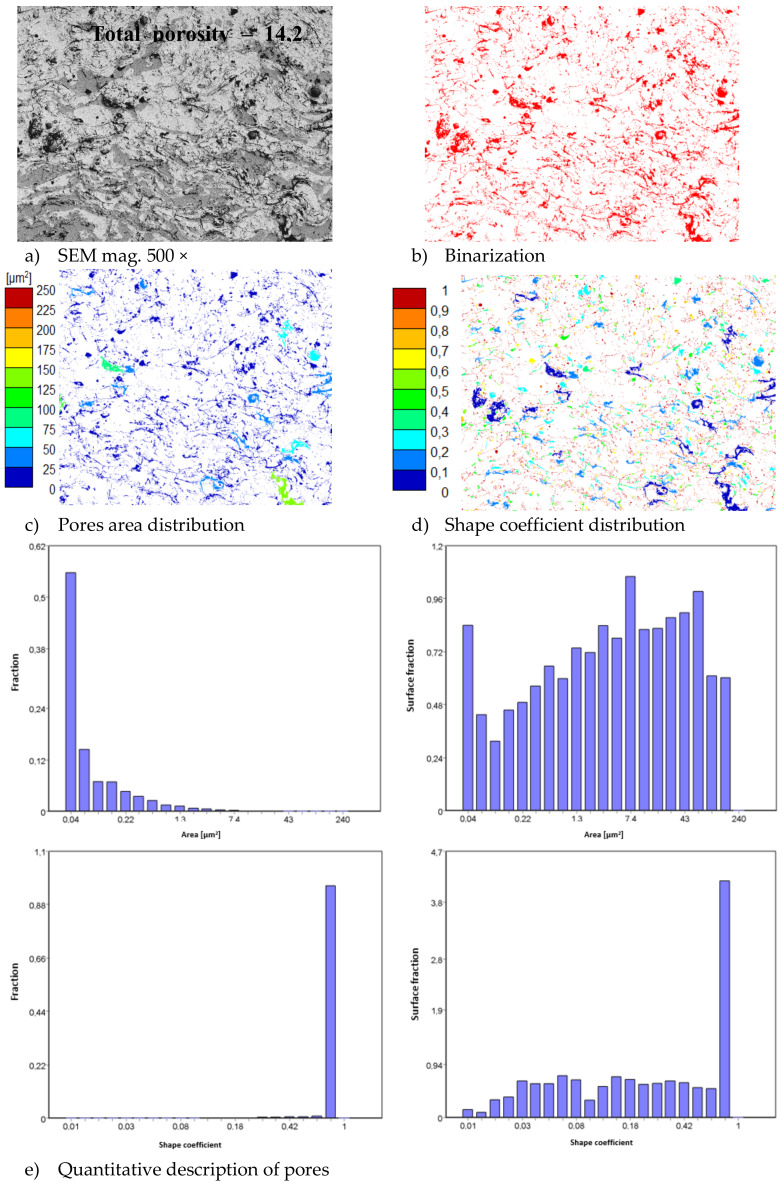
The image analysis results describe pores and voids morphology from quantitative and qualitative points of view for the DCL La_2_Zr_2_O_7_/50% La_2_Zr_2_O_7_ + 50% 8YSZ/8YSZ system: (**a**) SEM image, (**b**) binarization of pores and cracks, (**c**) visual evaluation of pores area distribution, (**d**) visual evaluation of pores shape coefficient distribution, (**e**) quantitative evaluation of pores area and shape coefficient distribution.

**Figure 9 materials-13-03242-f009:**
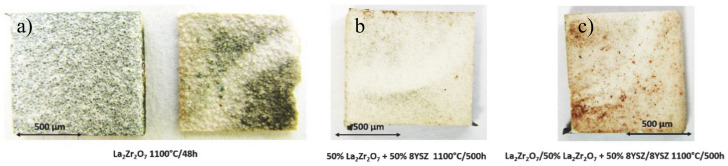
Top view of TBC specimens after the end of the oxidation test: (**a**) monolayered La_2_Zr_2_O_7_, (**b**) composite 50% La_2_Zr_2_O_7_ + 50% 8YSZ, and (**c**) DCL La_2_Zr_2_O_7_/50% La_2_Zr_2_O_7_ + 50% 8YSZ/8YSZ.

**Figure 10 materials-13-03242-f010:**
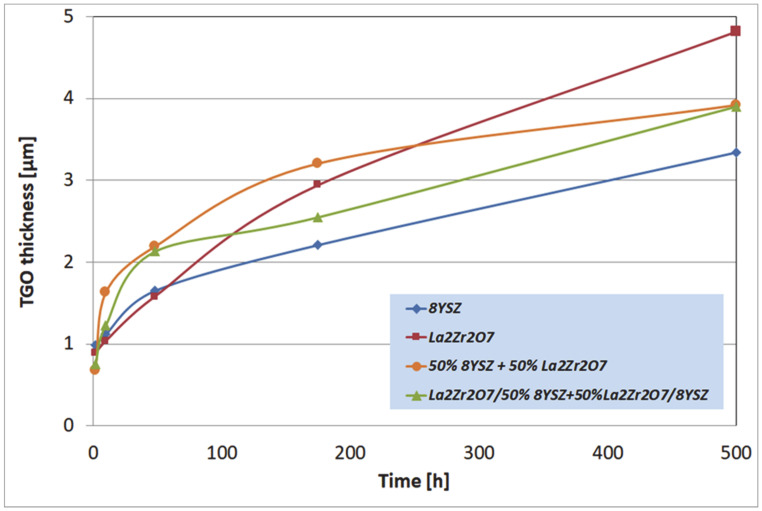
The thickness of the thermally grown oxide (TGO) zone as a function of oxidation time.

**Figure 11 materials-13-03242-f011:**
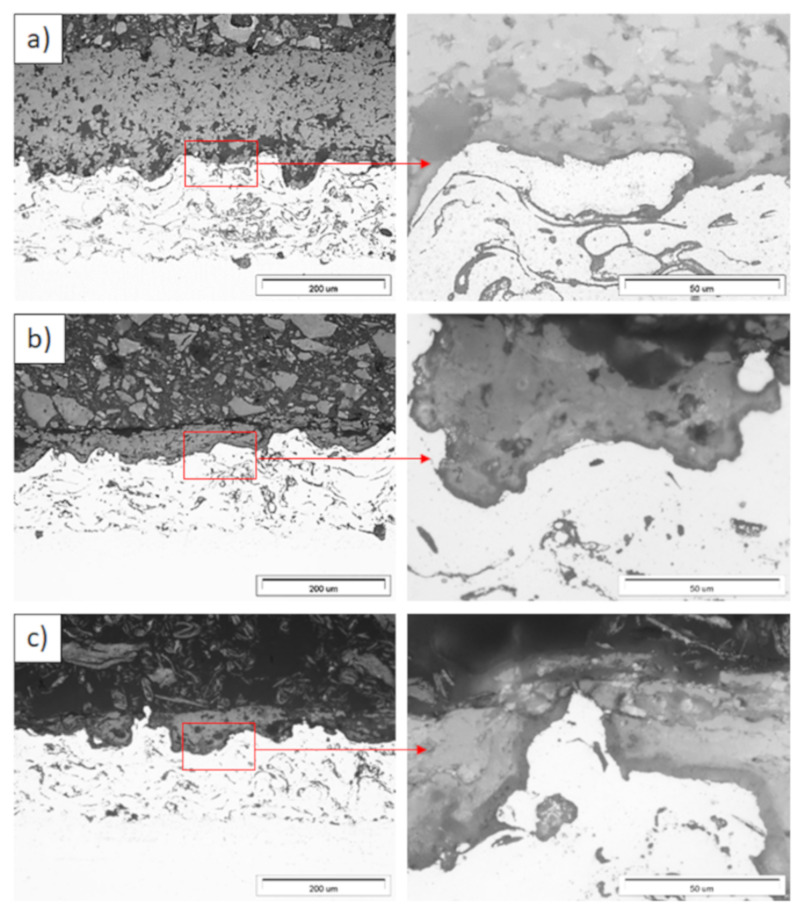
Ceramic layer/bond-coat interfaces of the TBC systems as a function of oxidation time at a temperature of 1100 °C—monolayered La_2_Zr_2_O_7_ TBC (light microscopy): (**a**) 2 h, (**b**) 48 h, and (**c**) 175 h.

**Figure 12 materials-13-03242-f012:**
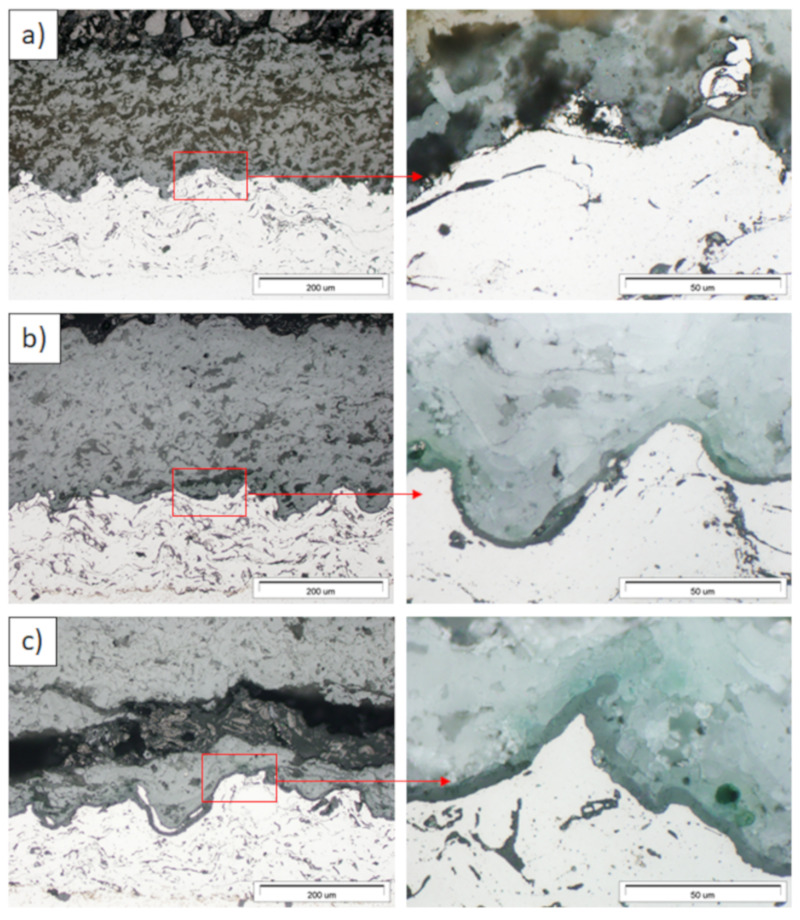
Ceramic layer/bond-coat interfaces of the TBC systems as a function of oxidation time at a temperature of 1100 °C—composite 50% La_2_Zr_2_O_7_ + 50% 8YSZ TBC (light microscopy): (**a**) 2 h, (**b**) 48 h, (**c**) 175 h.

**Figure 13 materials-13-03242-f013:**
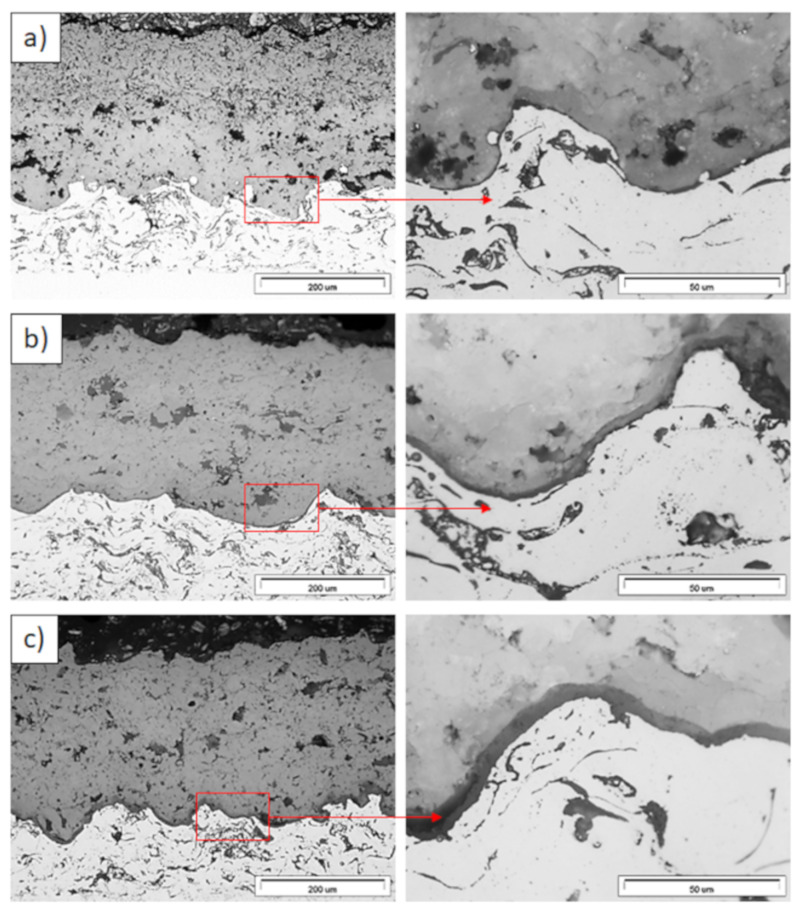
Ceramic layer/bond-coat interfaces of TBC systems as a function of oxidation time at a temperature of 1100 °C—DCL La_2_Zr_2_O_7_/50% La_2_Zr_2_O_7_ + 50% 8YSZ/8YSZ TBC (light microscopy): (**a**) 2 h, (**b**) 48 h, and (**c**) 175 h.

**Figure 14 materials-13-03242-f014:**
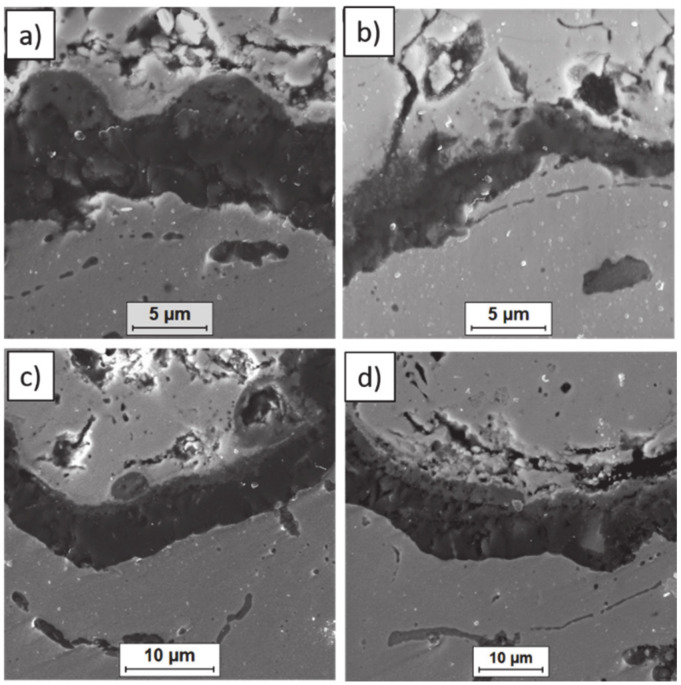
Ceramic layer/bond-coat interfaces of the TBC systems after 500 h of oxidation at a temperature of 1100 °C—monolayered La_2_Zr_2_O_7_ TBC (SEM): (**a**,**b**) zone of the bond-coat top and (**c**,**d**) zone of the bond-coat bottom.

**Figure 15 materials-13-03242-f015:**
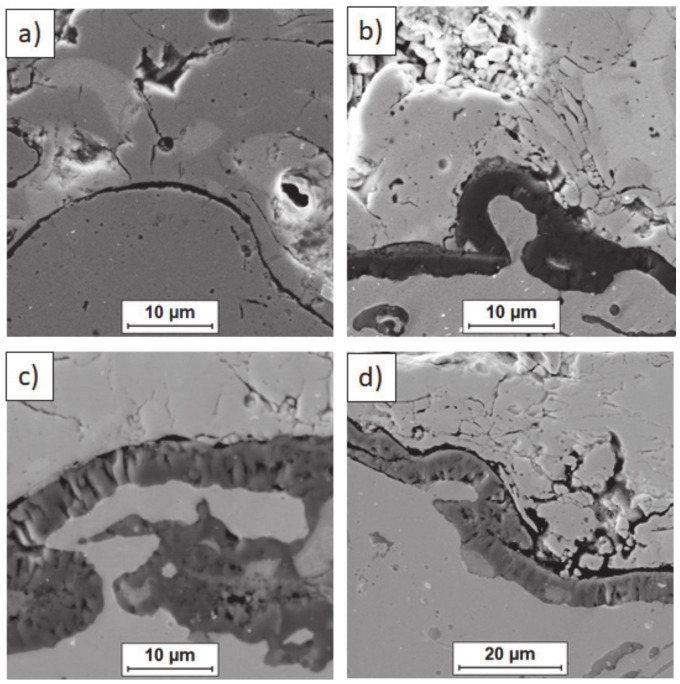
Ceramic layer/bond-coat interfaces of TBC systems after 500 h of oxidation at a temperature of 1100 °C—composite 50% La_2_Zr_2_O_7_ + 50% 8YSZ TBC (SEM): (**a**,**b**) zone of the bond-coat top and (**c**,**d**) zone of the bond-coat bottom.

**Figure 16 materials-13-03242-f016:**
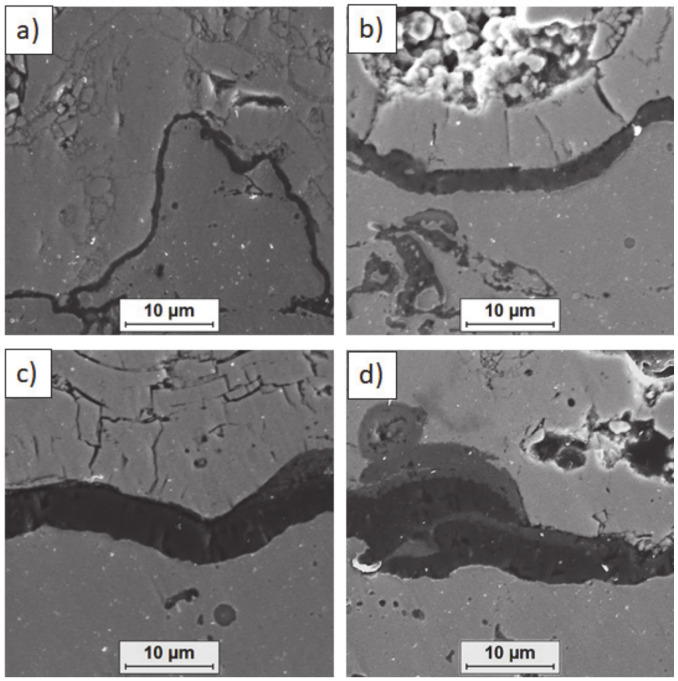
Ceramic layer/bond-coat interfaces of the TBC systems after 500 h of oxidation at a temperature of 1100 °C—DCL La_2_Zr_2_O_7_/50% La_2_Zr_2_O_7_ + 50% 8YSZ/8YSZ TBC (SEM): (**a**,**b**) zone of the bond-coat top and (**c**,**d**) zone of the bond-coat bottom.

**Figure 17 materials-13-03242-f017:**
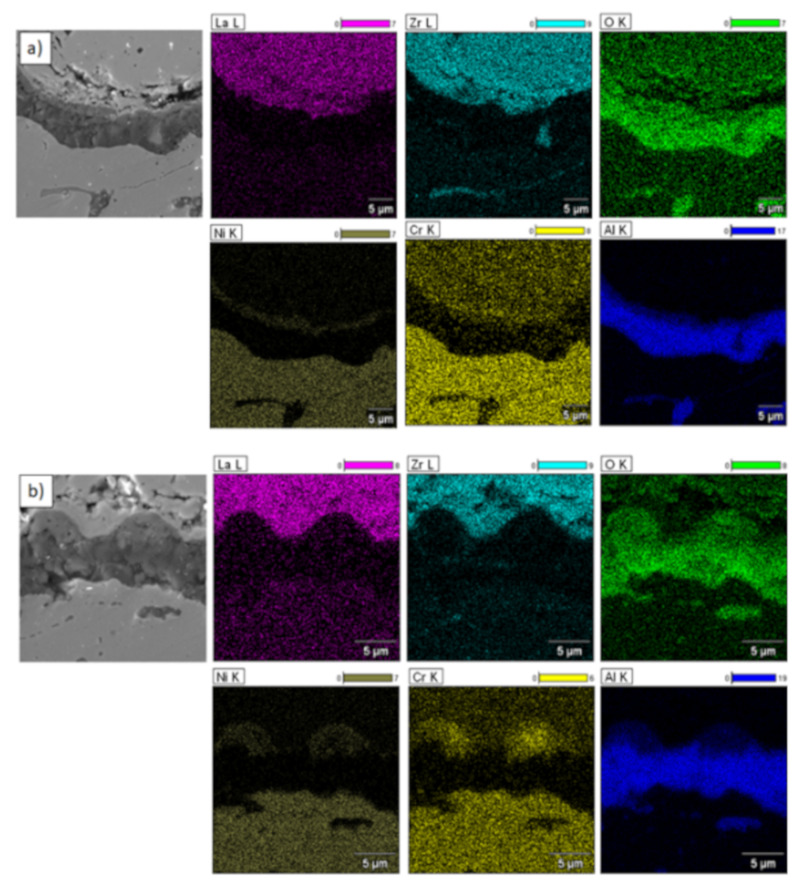
Element distribution in the TGO zone after 500 h of oxidation at a temperature of 1100 °C—monolayered La_2_Zr_2_O_7_ TBC (SEM): (**a**) zone of the bond-coat bottom and (**b**) zone of the bond-coat top.

**Figure 18 materials-13-03242-f018:**
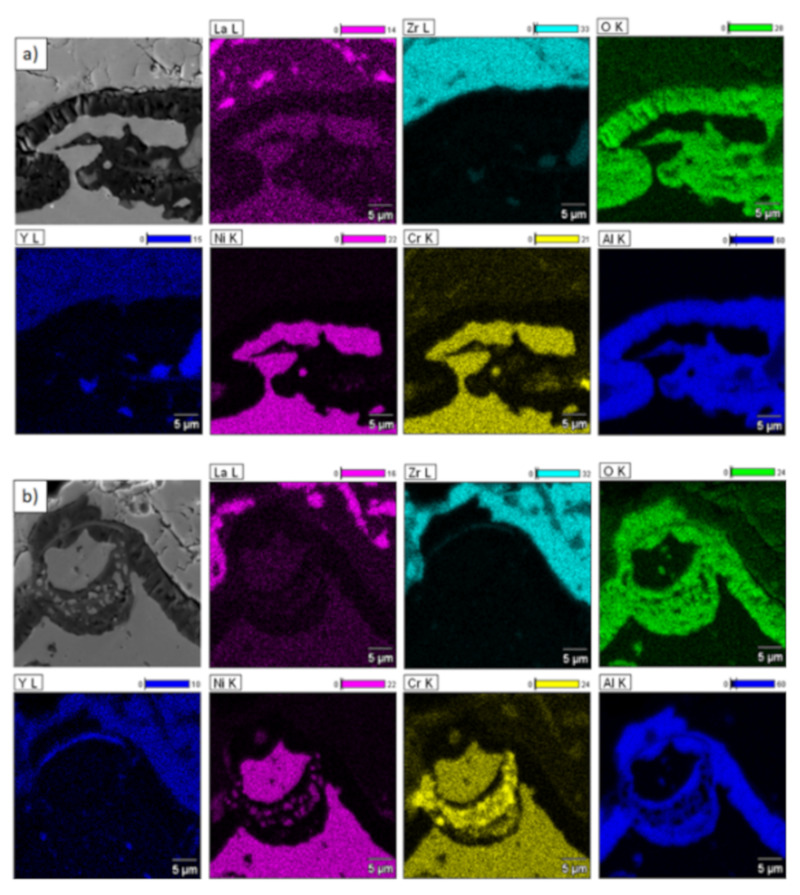
Element distribution in TGO zone after 500 h of oxidation at a temperature of 1100 °C—composite 50% La_2_Zr_2_O_7_ + 50% 8YSZ TBC (SEM): (**a**) zone of the bond-coat bottom and (**b**) zone of the bond-coat top.

**Figure 19 materials-13-03242-f019:**
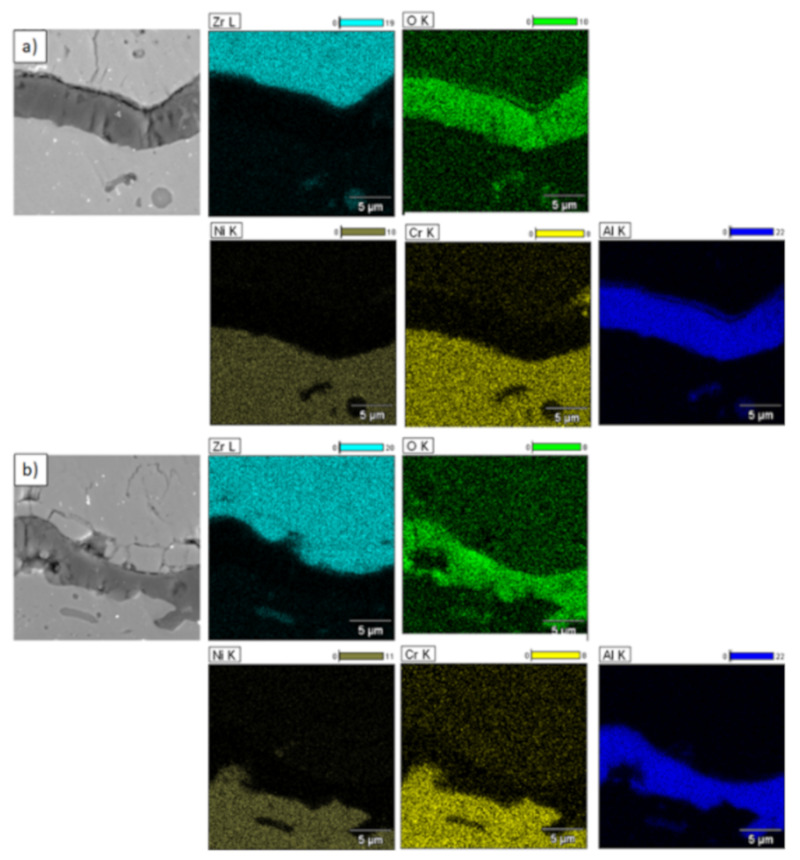
Element distribution in TGO zone after 500h of oxidation at a temperature of 1100 °C—DCL La_2_Zr_2_O_7_/50% La_2_Zr_2_O_7_ + 50% 8YSZ/8YSZ TBC (SEM): (**a**) zone of the bond-coat bottom and (**b**) zone of the bond-coat top.

**Figure 20 materials-13-03242-f020:**
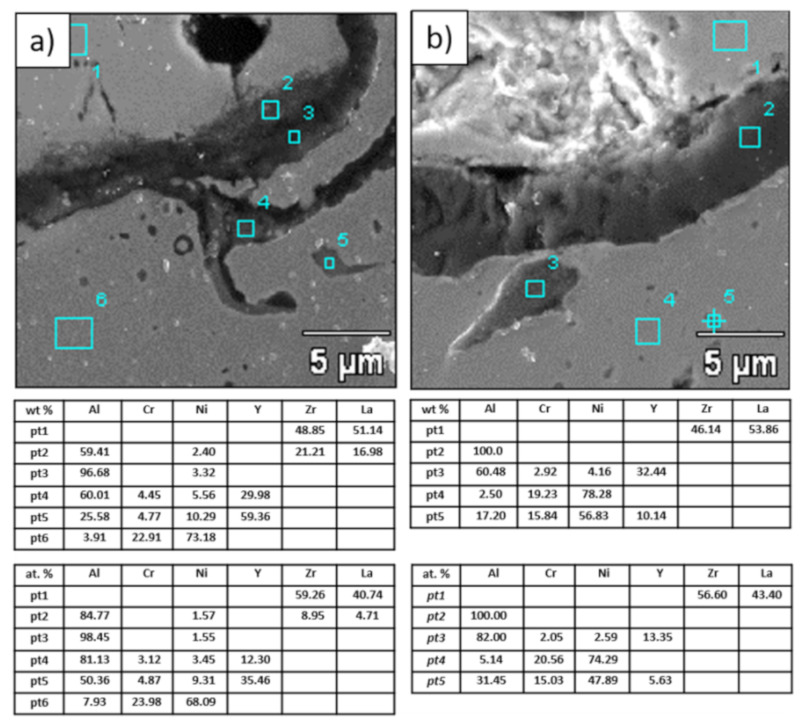
Analysis of chemical composition in micro-areas of TGO zone after 500 h of oxidation at a temperature of 1100 °C—monolayered La_2_Zr_2_O_7_ TBC (SEM): (**a**) zone of the bond-coat bottom and (**b**) zone of the bond-coat top.

**Figure 21 materials-13-03242-f021:**
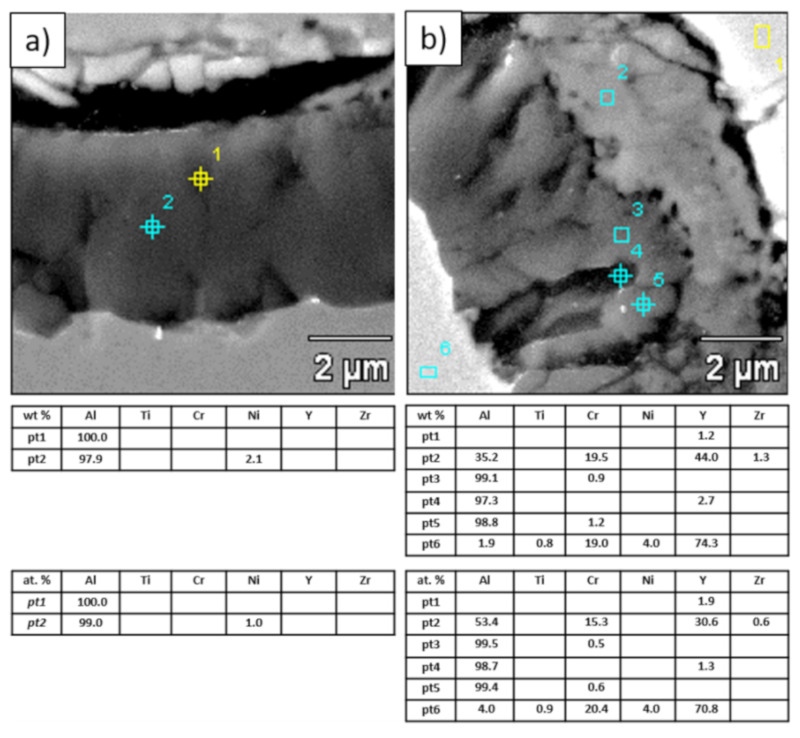
Analysis of chemical composition in micro-areas of the TGO zone after 500 h of oxidation at a temperature of 1100 °C—composite 50% La_2_Zr_2_O_7_ + 50% 8YSZ TBC (SEM): (**a**) zone of the bond-coat bottom and (**b**) zone of the bond-coat top.

**Figure 22 materials-13-03242-f022:**
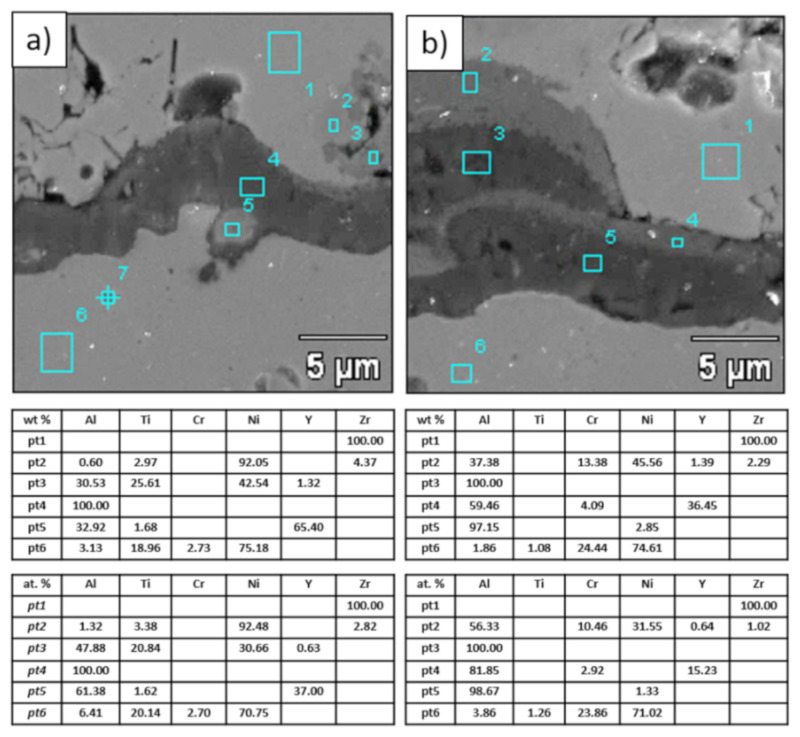
Analysis of chemical composition in micro-areas of TGO zone after 500 h of oxidation at a temperature of 1100 °C—DCL La_2_Zr_2_O_7_/50% La_2_Zr_2_O_7_ + 50% 8YSZ/8YSZ TBC (SEM): (**a**) zone of the bond-coat bottom and (**b**) zone of the bond-coat top.

**Figure 23 materials-13-03242-f023:**
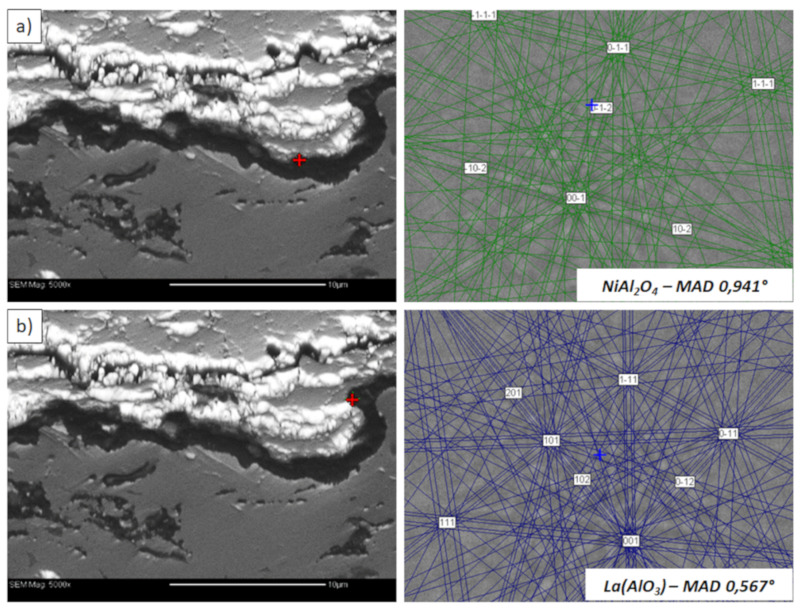
Analysis of phase composition in micro-areas of the TGO zone after 500 h of oxidation at a temperature of 1100 °C by EBSD—La_2_Zr_2_O_7_/TGO interface: (**a**) place of NiAl_2_O_4_ detection, (**b**) place of La(AlO_3_) detection.

**Figure 24 materials-13-03242-f024:**
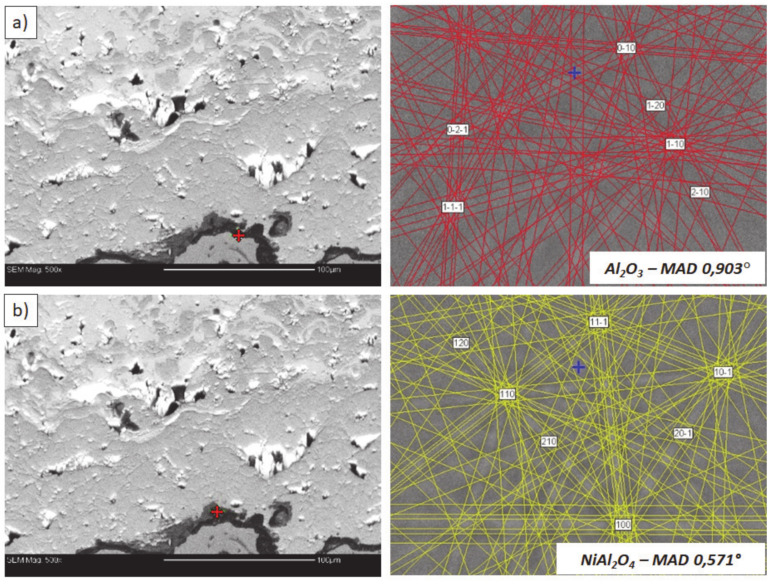
Analysis of phase composition in micro-areas of the TGO zone after 500 h of oxidation at a temperature of 1100 °C by EBSD—8YSZ/TGO interface: (**a**) place of Al_2_O_3_ detection, (**b**) place of NiAl_2_O_4_ detection.

**Figure 25 materials-13-03242-f025:**
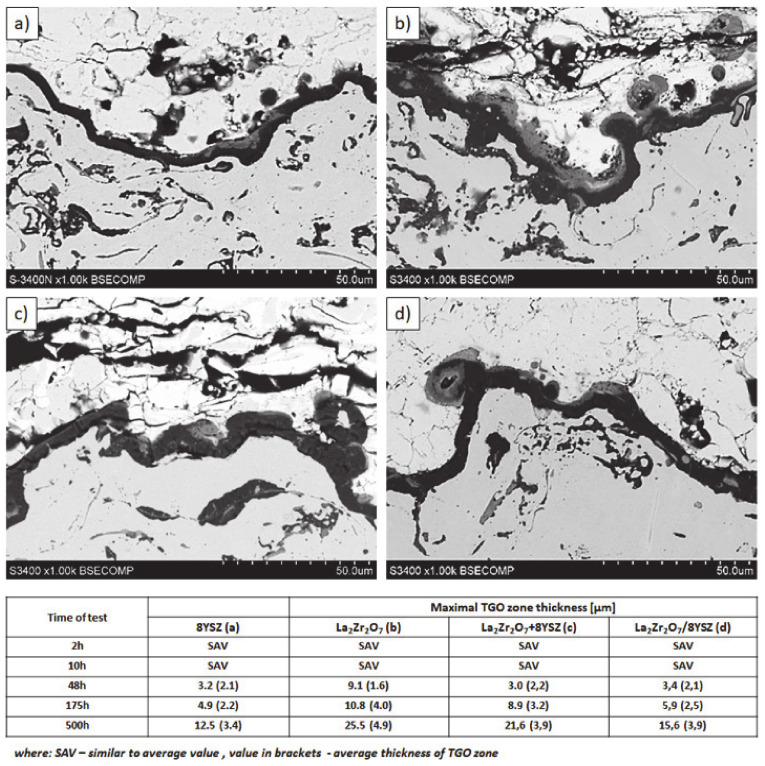
Quantitative description of the TGO zone thickness as a function of oxidation time: (**a**) 8YSZ monolayered TBC, (**b**) monolayered La_2_Zr_2_O_7_, (**c**) composite 50% La_2_Zr_2_O_7_ + 50% 8YSZ, and (**d**) DCL La_2_Zr_2_O_7_/50% La_2_Zr_2_O_7_ + 50% 8YSZ/8YSZ.

**Figure 26 materials-13-03242-f026:**
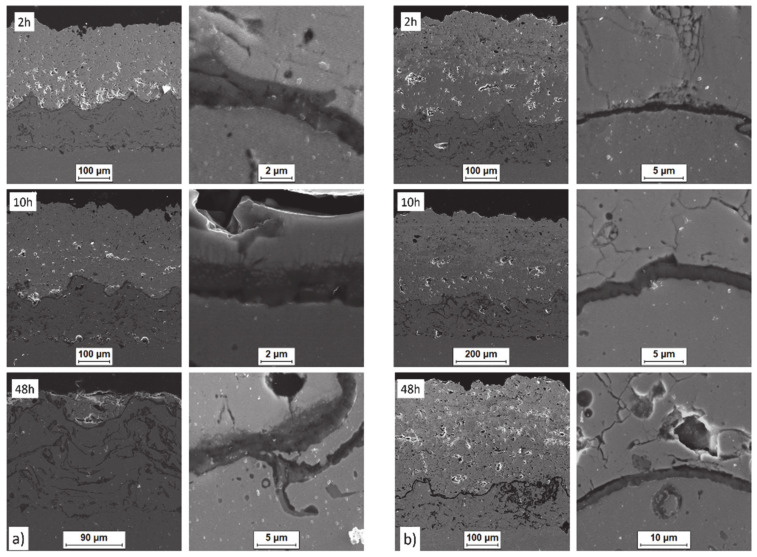
Crack nucleation and growth in the ceramic top layer and, phenomena during TGO zone formation as a function of time of oxidation: (**a**) monolayered La_2_Zr_2_O_7_ and (**b**) DCL La_2_Zr_2_O_7_/50% La_2_Zr_2_O_7_ + 50% 8YSZ/8YSZ.

**Figure 27 materials-13-03242-f027:**
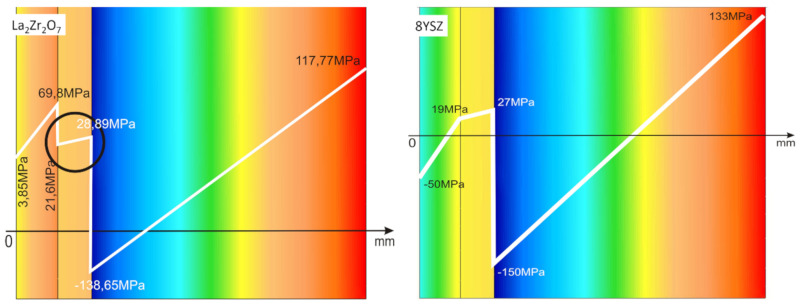
FEM (finite elements modelling) analysis of stress distribution in the area of La2Zr2O7/TGO and 8YSZ/TGO interfaces.

**Figure 28 materials-13-03242-f028:**
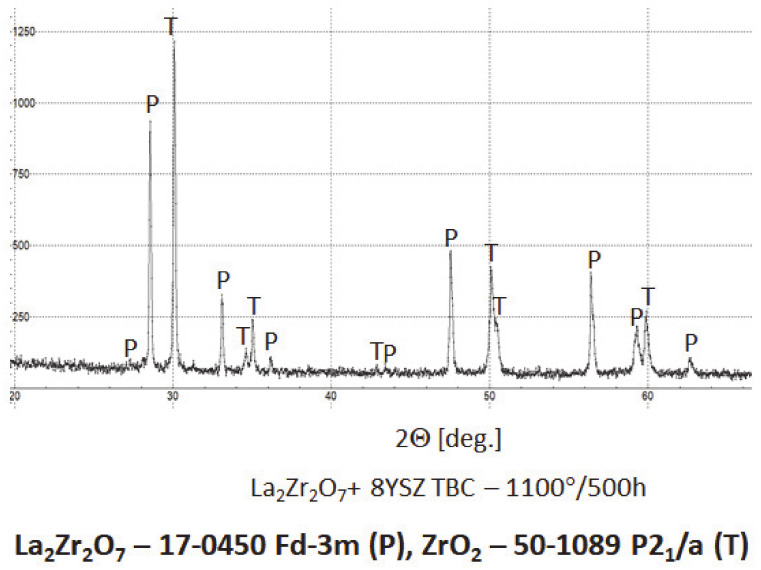
Phase composition of the ceramic top-coat of 0% La_2_Zr_2_O_7_ + 50% 8YSZ TBC composite after oxidation test at 1100 °C with 500 h of exposure.

**Figure 29 materials-13-03242-f029:**
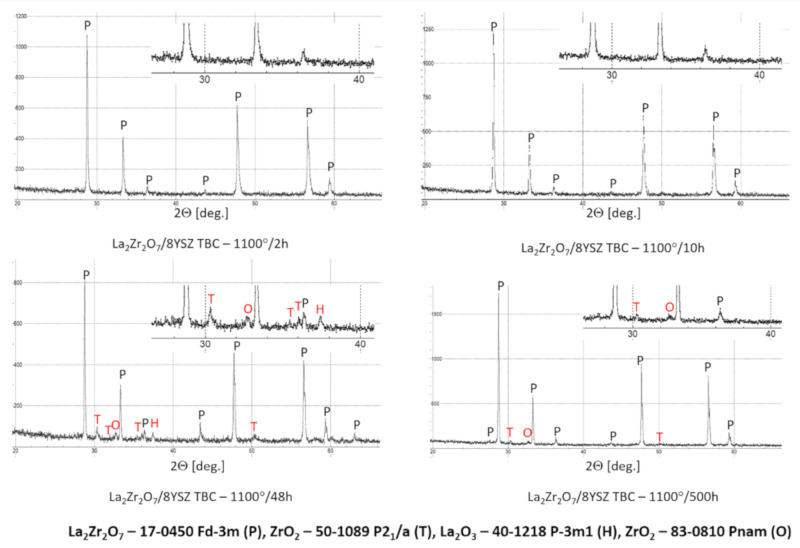
Phase composition of the ceramic top-coat of DCL La_2_Zr_2_O_7_/50% La_2_Zr_2_O_7_ + 50% 8YSZ/8YSZ TBC after oxidation test at 1100 °C with 500 h of exposure.

**Figure 30 materials-13-03242-f030:**
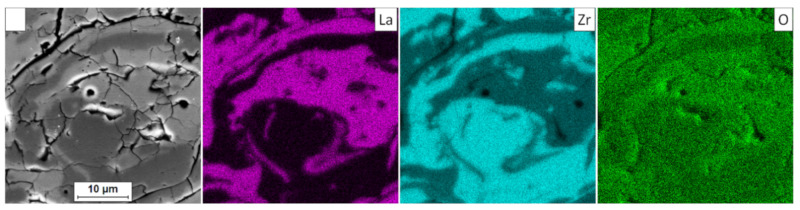
Element distribution in the area of the La_2_Zr_2_O_7_/8YSZ interface.

**Figure 31 materials-13-03242-f031:**
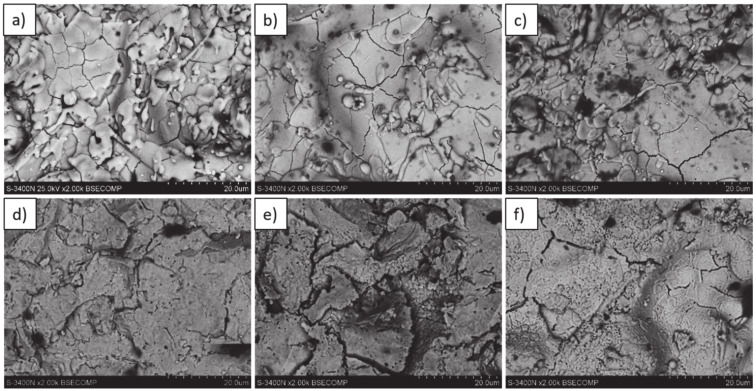
Top-coat topography in as-sprayed conditions (top) and after oxidation test (bottom): (**a**,**d**) monolayered La_2_Zr_2_O_7_, (**b**,**e**) composite 50% La_2_Zr_2_O_7_ + 50% 8YSZ, and (**c**,**f**) DCL La_2_Zr_2_O_7_/50% La_2_Zr_2_O_7_ + 50% 8YSZ/8YSZ.

**Table 1 materials-13-03242-t001:** Chemical composition of the feedstock powders.

(wt.%)	8YSZ	La_2_Zr_2_O_7_
Zr	Bal.	Bal.
La	-	59.1 + 0.90
Y	6.10 + 0.10	0.009 + 0.001
Al	0.17 + 0.008	0.092 + 0.001
Si	<0.10	<0.10
Cu	<0.01	<0.01
Ti	<0.067 + 0.003	<0.011 + 0.003
S	0.004 + 0.0004	0.001 + 0.0002
C	0.018 + 0.002	0.002 + 0.002
O	12.9 + 0.08	10.1 + 0.08
N	101 ppm + 10	434 ppm + 10

**Table 2 materials-13-03242-t002:** Technological parameters of powders used for plasma spraying of thermal barrier coating (TBC) systems.

Technological Parameters	8YSZ	La_2_Zr_2_O_7_
Density (g/cm^3^)	5.89 ± 0.12	5.78 ± 0.17
Bulk density by funnel method (g/cm^3^)	2.06 ± 0.03	1.23 ± 0.02
Bulk density by Scott method (g/cm^3^)	2.12 ± 0.04	1.30 ± 0.02
Flowability (s/50 g)	45 ± 2	49 ± 2

**Table 3 materials-13-03242-t003:** Parameters of the atmospheric plasma spraying process.

Parameters	Powder Composition			
	8YSZ	La_2_Zr_2_O_7_	La_2_Zr_2_O_7_/8YSZ	La_2_Zr_2_O_7_ + 8YSZ
Type of plasma gun	F4MB	F4MB	F4MB	F4MB
Argon (L/min)	40	40	40	40
Hydrogen (L/min)	10	10	10	10
Current (A)	600	600	600	600
Arc voltage (V)	59.3–60.3	61.6–62.3	60.6–62.1	60.9–62.8
Torchpower (W)	36.5–35.9	37.2–38.1	37.3–38.2	37.7–38.9
Feed (mm/s)	10	10	10	10
Distance (mm)	100	100	100	100
Number of cycles	20	20	30	30

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
