# Peer review of "Oxidation Behavior of the Monolayered La2Zr2O7, Composite La2Zr2O7 + 8YSZ, and Double-Ceramic Layered La2Zr2O7/La2Zr2O7 + 8YSZ/8YSZ Thermal Barrier Coatings"

_materials, 2020, doi:10.3390/ma13143242_

Round 1
Reviewer 1 Report
Jasik et al. describe oxidation behavior and phase stability of La2Zr2O7 and YSZ containing monolayer, composite and double-layer thermal barrier coating systems after 500h treatment at 1100°C. This is a research topic that has been explored for more than 15 years already and the same coating systems have been surely investigated and reported in the literature even at higher temperatures, under thermal gradients and cyclic conditions, at different atmospheres, etc. Therefore, given the previously published work in the field, this reviewer cannot recommend the manuscript for publication in Coatings on the grounds of lack of novelty. Furthermore, the reviewer finds the English level of the manuscript not acceptable for publication. There are even sentences repeating themselves in a row (page 2, line 63-65) and these kinds of mistakes should have been corrected by authors before the review process.
Author Response
The Authors thank you for the critical review, but they do not disagree about its substantive content.
First of all, the manuscript presents research on three types of TBC systems, i.e. monolayered, composite and FGS, i.e. combinations of DCL and composite multilayers.
One can agree to the extent that monolayered coatings based on La2Zr2O7 have already been tested, while one cannot agree on composite type coatings (both other tested systems). The number of publications is minimal in this area.
In addition, the following article applies not only to phenomena in the TGO zone, but above all to phase stability in the system of two structural components in composite coatings. This is the latest research direction in the field of TBC coatings. Just look at our two latest publications in this area, i.e.
- G. Moskal, A. Jasik, S. Jucha, Mikuśkiewicz M., Thermal resistance determination of Sm2Zr2O7 + 8YSZ composite type of TBC, Applied Surface Science 515 (2020) 145998, DOI: 10.1016 / j.apsusc.2020.145998
- G. Moskal, S. Jucha, M. Mikuśkiewicz, D. Migas, A. Jasik, Atypical decomposition processes of Sm2Zr2O7 + 8YSZ dual-phase TBCs during hot corrosion, Corrosion Science (2020) 108681,
DOI: 10.1016 / j.corsci.2020.108681
English has been corrected.
Reviewer 2 Report
This paper describes oxidation behaviour of monolayered La2Zr2O7, composite La2Zr2O7+8YSZ, and double-ceramic layered La2Zr2O7/La2Zr2O7+8YSZ/8YSZ thermal barrier coatings (TBC). In particular, the behaviour under air environment at 1100°C for 500 hours exposure was investigated with the aim of characterizing the oxidation induced degradation processes in different TBC systems. Phase stability due to temperature effect in ceramic top coats was also investigated. Some numerical simulation of monolayered TBC 8YSZ and 22 La2Zr2O7 systems was carried out in order to simulate the oxidation induced stress distribution.
The subject of the paper is original and seems overall coherent with the aim of the journal. The English is acceptable but can be improved. The presentation is overall clear, though some points must be properly clarified.
The experimental analysis is very accurate and conclusive. The numerical simulations need to be properly introduced and discussed. Overall, the work needs being properly introduced and situated with respect to the state of the art.
The following comments should be properly taken into account before paper publication (minor but mandatory revision is needed):
- The Introduction section does not state clearly what is the novelty of the paper with respect to the state of the art. This must be properly and clearly addressed.
- The features of the numerical model are not sufficiently introduced and presented. Why a model is needed? How the model is built? Which are the outputs of the model and the main results? This is very poorly addressed. The (beneficial?) contribution from numerical simulations is not sufficiently made clear in the discussion and in the conclusion section.
- The conclusion section should report the paper findings in a more robust fashion. The results of numerical simulations are not properly reported in the conclusive discussion, one is left with the idea that they are almost useless.
Author Response
Thank you very much for the substantive review.
According to the suggestions, the English language has been corrected.
Changes were introduced in the scope of numerical tests and their intermperativeness.
Description explaining the area of novelty in conducted research was introduced.
Reviewer 3 Report
The paper presents important aspect of novel TBC top coats. The work is of interest as new materials are being sought to replace the YSZ system that has been used for decades. However, the paper needs extensive editing of English to make it easier for the reader to follow. In addition, a few aspects of the work needs clarification:
- The literature in the introduction requires a bit more information. There are several papers in 'Coatings' Journal that could be referenced. Please have a look at the literature and add those relevant references.
- The difference in the thickness of the three coatings studied was not discussed. This needs to be discussed and a method to normalise the data must be presented.
- The information on FEA is limited. Please provide information on the model used. If this has been taken from literature then please provide relevant literature. If the model was developed for the paper then further information on the model parameters is required.
- The quality of presentation of the graphs needs to improve. There are several plots where the text cannot be understood.

Author Response
The authors thank you for the substantive review of the article.
Literature data has been modified.
English has been corrected.
Information on numerical simulations has been introduced.
The effect of thickness was determined and analyzed.
The quality of drawings and charts has been improved.
Round 2
Reviewer 1 Report
- The manuscript is more readable after English corrections however this doesn’t change the content of the paper
- As listed in the reference list (#10), in 2004, Vaßen et al proposed and showed that a double layer system (pyrohlore+ysz) performs better than a single layer (pyrochlore) or a YSZ system up to temperatures 1450°C. In the past 16 years, the same result was shown for several different pyrochlore compositions, different deposition processes, composite or graded structures. Manuscript of Jasik et al shows the same ultimate result for single and double layer composite systems.
- It is not understandable why the La2zr2o7 double-layer phase composition is only indexed with the zirconia phase in figure 4c.
- The authors characterize tgo phase compositions and as Leckie showed in 2005 (acta mater. 53(11), 3281-3292), they find a porous LaAlO3 layer at the topcoat-tgo interface when La2Zr2O7 is in contact with alumina.
- The authors compare the tgo thickness of different systems without comparing porosity levels or layer thicknesses of the topcoat layers and without giving error bars. Considering that pyrochores and ysz have different melting points (roughly YSZ=2700C, La2Zr2O7=2280C) it is well expected that the investigated layers have different porosities as they were deposited at the same deposition conditions.
- Finally the authors find out that “the system of La2Zr2O7 + 8YSZ type is thermo-chemically stable in a condition of high-temperature oxidation”. This is consistent with many studies in the literature. In fact, according to the phase diagram, La2Zr2O7 is a stable oxide till it melts, so why it should undergo a thermal decomposition? After longer times at high temperatures, metastable zirconia can be observed in the la2zr2o7 coatings but this is a phenomenon related to evaporation of La2O3 during plasma spraying.
After all, again, due to lack of novelty, the reviewer suggests rejection.
Author Response
Thank You for Your comments.
1.
In the article of Robert Vassen, the DCL system with sharp boundary between 8YSZ and La2Zr2O7, was analyzed. Generally the main part of publications about DCL systems is related to coatings of this type. In our investigations, we present the coating with additional sublayer consists the mixture of 8YSZ phase and La zirconate. The information about these internal morphology are limited, such as about composite dual phase TBC systems. For this type of TBC systems the most important factor determining their durability, it is CTE mismatch, and interphase instability between 8YSZ and Ln zirconates.
Our publications e.g.
https://doi.org/10.1016/j.corsci.2020.108681
showed clearly that dual phase systems of 8YSZ-Ln2Zr2O7 type exhibit strong tendency to interaction.
Those observations are confirmed by:
https://doi.org/10.1016/j.matlet.2018.09.088
In the case of La2Zr2O7-8YSZ system the publication:
http://dx.doi.org/10.1016/j.ceramint.2015.10.046
showed that the phase stability of tetragonal YSZ was decreased with the increase in sintering temperature along with the increase in volume fraction o LZ in LZ/YSZ composites. The addition of LZ in YSZ resulted in the phase transformation from tetragonal YSZ to monoclinic YSZ at high temperatures. The fraction of cubic phase was also found to increase with the increase in sintering time and temperature.
In our opinion there is not correct assumption, that all type of lanthanides showed the same behavior in condition of oxidation, hot corrosion or thermal shock.
2.
The figure 4c was corrected. It was manual mistake.
3.
The publication of Lackie is related to GdAlO3 phase formation not LaAlO3.
4.
The quantitative and qualitative characterization of coatings porosity and cracks architecture was added. The thickness of composite and FGS coatings is higher, but simultaneously the porosity is much higher.
5. How we showed in point 1, the thermal stability of La2Zr2O7+8YSZ system is not clearly described
(see: http://dx.doi.org/10.1016/j.ceramint.2015.10.046) and conclusion about phase instabilities.
The Reviewer presents opinion about thermal stability only of La2Zr2O7 phase, and all presented assumption are true. But in presented manuscript, we analyzed the dual phase system of La2Zr2O7+8YSZ type. Our investigations of Sm2Zr2O7+8YSZ, Nd2Zr2O7+8YSZ, and Gd2Zr2O7+8YSZ, revealed that there is strong tendency to formation of fluorite type of zirconates both in pure oxidation (in higher temperature) as well as in environment of liquid sulphate salts.
This investigations revealed that in this condition, it can be expected that both the La2Zr2O7 and 8YSZ compounds are thermally stable, contrary to others lanthanides such as Nd, Sm or Gd. The basic reason of this behavior is related to cation size differences, what was confirmed in investigations of CMAS corrosion:
https://ceramics.onlinelibrary.wiley.com/doi/full/10.1111/jace.16694
Reviewer 3 Report
Thank you for addressing the comments. The paper is in a much better shape. However, this still requires some editorial work in terms of English language usage. I would recommend that you review the paper carefully and check for grammatical errors and resubmit the paper. The technical content is okay.
Author Response
Thank You for Your suggestions. The english language was corrected by native speaker in accordance to Your suggestion.